# ON THE UNIVERSAL APPROXIMABILITY AND COMPLEXITY BOUNDS OF DEEP LEARNING IN HYBRID QUANTUM-CLASSICAL COMPUTING

## ABSTRACT

With the continuously increasing number of quantum bits in quantum computers, there are growing interests in exploring applications that can harvest the power of them. Recently, several attempts were made to implement neural networks, known to be computationally intensive, in hybrid quantum-classical scheme computing. While encouraging results are shown, two fundamental questions need to be answered: (1) whether neural networks in hybrid quantum-classical computing can leverage quantum power and meanwhile approximate any function within a given error bound, i.e., universal approximability; (2) how do these neural networks compare with ones on a classical computer in terms of representation power? This work sheds light on these two questions from a theoretical perspective.

## 1 INTRODUCTION

Quantum computing has been rapidly evolving (e.g., IBM (2020) recently announced to debut quantum computer with 1,121 quantum bits (qubits) in 2023), but the development of quantum applications is far behind; in particular, it is still unclear what and how applications can take quantum advantages. Deep learning, one of the most prevalent applications, is well known to be computation-intensive and therefore their backbone task, neural networks, is regarded as an important task to potentially take quantum advantages. Recent works (Francesco et al., 2019; Tacchino et al., 2020; Jiang et al., 2020) have demonstrated that the shallow neural networks with limited functions can be directly implemented on quantum computers without interfering with classical computers, but as pointed by Broughton et al. (2020), the near-term Noisy Intermediate-Scale Quantum (NISQ) can hardly disentangle and generalize data in general applications, using quantum computers alone. This year, Google (2020) has put forward a library for hybrid quantum-classical neural networks, which attracts attention from both industry and academia to accelerate quantum deep learning.

In a hybrid quantum-classical computing scheme, quantum computers act as hardware accelerators, working together with classical computers, to speedup the neural network computation. The incorporation of classical computers is promising to conduct operations that are hard or costly to be implemented on quantum computers; however, it brings high data communication costs at the interface between quantum and classical computers. Therefore, instead of contiguous communication during execution, a better practice is a "prologue-acceleration-epilogue" scheme: the classical computer prepares data and post-processes data at prologue and epilogue, while only the quantum computer is active during the acceleration process for the main computations. Without explicit explanation, "hybrid model" refers to the prologue-acceleration-epilogue scheme in the rest of the paper.

In a classical computing scheme, the universal approximability, i.e., the ability to approximate a wide class of functions with arbitrary small error, and the complexity bounds of different types of neural networks have been well studied (Cybenko, 1989; Hornik et al., 1989; Mhaskar & Micchelli, 1992; Sonoda & Murata, 2017; Yarotsky, 2017; Ding et al., 2019; Wang et al., 2019; Fan et al., 2020). However, due to the differences in computing paradigms, not all types of neural networks can be directly implemented on quantum computers. As such, it is still unclear whether those can work with hybrid quantum-classical computing and still attain universal approximability. In addition, as quantum computing limits the types of computations to be handled, it is also unknown whether the

hybrid quantum-classical neural networks can take quantum advantage over the classical networks under the same accuracy. This work explores these questions from a theoretical perspective.

In this work, we first illustrate neural networks that are feasible in hybrid quantum-classical computing scheme. Then we use the method of bound-by-construction to demonstrate their universal approximability for a wide class of functions and the computation bounds, including network depth, qubit cost and gate cost, under a given error bound. In addition, compared with some of the lower complexity bounds for neural networks on classical computers, our established upper bounds are of lower asymptotic complexity, showing the potential of quantum advantage.

## 2 RELATED WORKS AND MOTIVATION

### 2.1 NEURAL NETWORKS IN QUANTUM COMPUTING

Although the research on neural networks in quantum computing can trace back to the 1990s (Kak, 1995; Purushothaman & Karayiannis, 1997; Ezhov & Ventura, 2000), but only recently, along with the revolution of quantum computers, the implementation of neural networks on actual quantum computer emerges (Francesco et al., 2019; Jiang et al., 2020; Bisarya et al., 2020). There are mainly three different directions to exploit the power of quantum computers: (1) applying the Quantum Random Access Memory (QRAM) (Blencowe, 2010); (2) employing pure quantum computers; (3) bridging different platforms for a hybrid quantum-classical computing (McClean et al., 2016).

Kerenidis et al. (2019) is a typical work to implement neural networks with QRAM. Using QRAM provides the highest flexibility, such as implementing non-linear functions using lookup tables. But QRAM itself has limitations: instead of using the widely applied superconducting qubits (Arute et al., 2019; IBM, 2016), QRAM needs the support of spin qubit (Veldhorst et al., 2015) to provide relatively long lifetime. To make the system practical, there is still a long way to go.

Alternatively, there are works which encode data to either qubits (Francesco et al., 2019) or qubit states (Jiang et al., 2020) and use superconducting-based quantum computers to run neural networks. These methods also have limitations: Due to the short decoherence times in current quantum computers, the condition statement is not supported, making it hard to implement some non-linear functions such as the most commonly used Rectified Linear Unit (ReLU). But the advantages are also obvious: (1) the designs can be directly evaluated on actual quantum computers; (2) little communication is needed between quantum and classical computers, which may otherwise be expensive.

Hybrid quantum-classical computing tries to address the limitations of QRAM and pure quantum computer based approaches. Broughton et al. (2020) establishes a computing paradigm where different neurons can be implemented on either quantum or classical computers. This brings the flexibility in implementing functions (e.g., ReLU), while at the same time, it calls for fast interfaces for massive data transfer between quantum and classical computers.

In this work, we focus on the hybrid quantum-classical computing scheme and follow the "prologue-acceleration-epilogue" computing scheme. It offers the flexibility of implementation and at the same time requires minimal quantum-classical data transfer, as demonstrated in Figure 2.

### 2.2 UNIVERSAL APPROXIMATION AND COMPLEXITY BOUND

Universal approximability of neural network indicates that for any given continuous function or a wide class of functions satisfying some constraints, and arbitrarily small error bound $\epsilon > 0$, there exists a neural network model which can approximate the function with no more than $\epsilon$ error. On classical computing, different types of neural networks have been proved to have universal approximability: multi-layer feedforward neural networks (Cybenko, 1989; Hornik et al., 1989); ReLU neural networks (Mhaskar & Micchelli, 1992; Sonoda & Murata, 2017; Yarotsky, 2017); quantized neural networks (Ding et al., 2019; Wang et al., 2019); and quadratic neural networks (Fan et al., 2020). In addition, many of these works also establishes complexity bounds in terms of the number of weights, number of layers, or number of neurons needed for approximation with error bound $\epsilon$.

When it comes to quantum computing, in recent years, Delgado (2018) demonstrated quantum circuit with an additional trainable diagonal matrix can approximate the given functions, and Schuld et al. (2020) shown that the Fourier-type sum-based quantum models can be universal function approximators if the quantum circuit is measured enough many times. Most recently, we are wit-

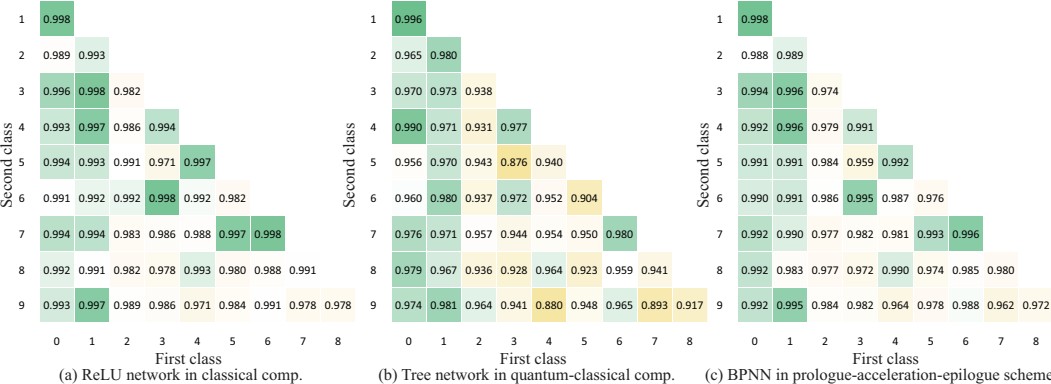

Figure 1: The test accuracy of neural networks on different computing platforms for pairwise classifiers in MNIST: (a) ReLU network in classical computing; (b) tree tensor network (Huggins et al., 2019) in hybrid quantum-classical computing; (c) the proposed BPNN in the prologue-acceleration-epilogue computing scheme (See Appendix B for detailed experimental setup).

nessing the exponentially increasing research works to exploit the high-parallelism provided by quantum computers to accelerate neural networks (Perdomo-Ortiz et al., 2018; Huggins et al., 2019; Cong et al., 2019; Kerenidis et al., 2019; Francesco et al., 2019; Bisarya et al., 2020; Broughton et al., 2020; Jiang et al., 2020; Tacchino et al., 2020; Xia & Kais, 2020). The existing works have demonstrated that the quantum neural network can achieve state-of-the-art accuracy for some tasks. For example, Figure 1 demonstrates the test accuracy comparison of different neural networks targeting pairwise classifiers in MNIST (LeCun et al., 1998)). The average accuracy gap between that in Figure 1(a) and Figure 1(c) is merely 0.5%. However, the penalty for achieving high-parallelism in quantum computing is the constrained computation types to be performed. As a result, neural networks designed for quantum computing has limited operations, indicating that the networks designed for classical computing may not be implemented on quantum computers. This raises a fundamental problem: whether the neural networks in quantum computing can attain universal approximability?

The failure to attain universal approximability will fundamentally hinder the neural networks in quantum computing being used in practice due to the significant accuracy loss for specific functions. Therefore, it is imminent to understand the expressivity of neural networks dedicated to quantum computers. Motivated by this, we prove that the neural networks in a hybrid quantum-classical computing can approximate a wide class of functions with arbitarily small error. We also establish complexity bounds, which gives practical insights in designing networks for quantum computing.

## 3 MAIN RESULTS

Figure 2 illustrates the adopted prologue-acceleration-epilogue computing scheme. It is a practical scheme with the small number of quantum-classical data transfer and the neural network designed for this scheme can achieve competitive accuracy against classical computing as demonstrated in Figure 1. In addition, the target computing scheme is a special case of that used in Tensorflow Quantum (Broughton et al., 2020); therefore, if we can prove that the neural networks designed for it have universal approximability, then the conclusion can be directly derived to Tensorflow Quantum. In this work, we follow the idea from (Yarotsky, 2017; Ding et al., 2019) by constructing a neural network (namely BPNN, see Section 4.1) in the prologue-acceleration-epilogue computing scheme (see Section 4.4) with bounded maximum error (see Section 4.3) for a wide class of functions (denoted as $\mathcal{F}_{d,n}$, see Section 4.2). In such a proof, the fundamental function to be implemented is the Taylor polynomial. In the next, we first state the main result of the Taylor polynomial of a function $f \in \mathcal{F}_{d,n}$ on the quantum computing (see Appendix A.5 for the formal version).

**Results 3.1.** *For any given function $f \in \mathcal{F}_{d,n}$ and an expansion point $\mathbf{k}$, its Taylor polynomial at point $\mathbf{k}$ can be implemented on the quantum computer, such that $(i)$ the network can exactly implements the Taylor polynomial, $(ii)$ the depth is $O(\log n)$, $(iii)$ the number of gates is $O(n^2 \log n \log d)$, $(iv)$ the number of qubits is $O(n^2 \log d)$.*

Here, we observe that since the Taylor function can be exactly implemented by the quantum computer, the complexities on depth, gates, and qubits are not related to error bound $\epsilon$. This will be the

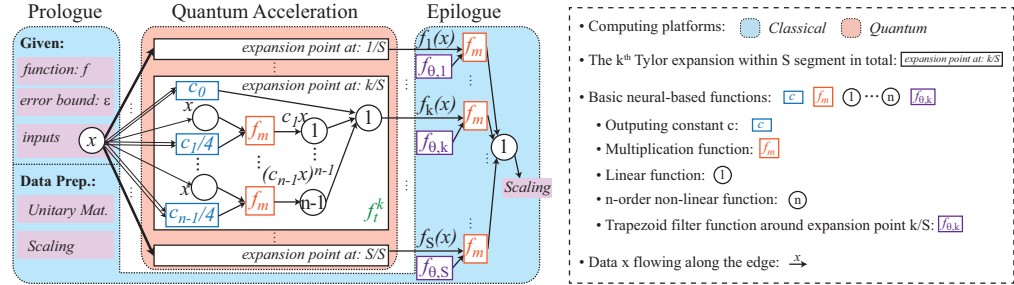

Figure 2: Illustration of the prologue-acceleration-epilogue computing scheme used in this work.

root cause that the upper bound of neural networks in hybrid quantum-classical computing scheme can approach to the lower bound of ones on classical computing (see the comparison at the end of this section). In addition, the classical computing cannot build such a system, because there are too many inputs, reaching up to $d^{n+1}$, which is infeasible for classical computing with exponentially increasing inputs. On the other hand, quantum computing can take advantage of encoding $n$ inputs to $\log n$ qubits, and therefore, it is feasible to implement such a network on a quantum computer.

The above result shows the ability of BPNN to exactly implement Taylor expansion at any point. Then, combined with the classical Prologue for quantum-state preparation and the classical Epilogue for accumulate results at all expansion points, we next state the main result of approximating a function $f \in \mathcal{F}_{d,n}$ on the prologue-acceleration-epilogue computing scheme as follows; the formal version can be found in Appendix A.6.

**Results 3.2.** *For any given function $f \in \mathcal{F}_{d,n}$, there is a binary polynomial neural network with a fixed structure that can be implemented in the hybrid quantum-classical computing scheme, such that $(i)$ the network can approximate $f$ with any error $\epsilon \in (0,1)$, $(ii)$ the overall depth is $O(1)$; $(iii)$ the number of quantum gates is $O\left((1/\epsilon)^{\frac{d}{n}}\right)$; $(iv)$ the number of qubits is $O\left((1/\epsilon)^{\frac{d}{n}}\right)$; $(v)$ the number of weights on classical computer is $O\left((1/\epsilon)^{\frac{d}{n}}\right)$.*

From the above result, we can see that the upper bounds on the depth and the number of weight/gates are of the same asymptotic complexity for both the quantum portion and classical portion in the hybrid computing system, which satisfies the constraint discussed in Section 4.1 to take full use of quantum computing . We further compare the complexity bounds between the BPNN on the hybrid quantum-classical computing scheme against the classical networks constructed for bound analysis.

**Comparison with the upper bounds for neural networks on classical computers:** To attain an approximation error $\epsilon$, Fan et al. (2020) demonstrates that the upper bound on the number of weights for unquantized quadratic network is $O(log(log(1/\epsilon)) \times (1/\epsilon)^{\frac{d}{n}})$, and Ding et al. (2019) demonstrates that the upper bound on the number of binary weights of the ReLU neural network is $O(log^2(1/\epsilon) \times (1/\epsilon)^{\frac{d}{n}})$. On the other hand, for the BPNN on hybrid quantum-classical computing, both the number of gates used in quantum acceleration and the weights used in classical prologue and epilogue are $O((1/\epsilon)^{\frac{d}{n}})$. Although BPNN has similar expressive power compared with the binary ReLU network and reduced expressive power compared with the unquantized quadratic network (due to the constraints on weight selection), the obtained upper bounds are of asymptotically lower complexity, which again shows the benefits of quantum computing for neural networks.

**Comparison with the lower bounds for neural networks on classical computers:** We further compare the lower bound of the number of weights/gates needed to attain an error bound $\epsilon$ on a classical computer. The only established result in the literature is for unquantized ReLU network (Yarotsky, 2017), which suggests that to attain an approximation error bound of $\epsilon$, the number of weights needed is at least $\Omega(\log^{-2p-1}(1/\epsilon) \times (1/\epsilon)^{d/n})$ with depth constraint of $O(\log^p(1/\epsilon))$ where $p$ is a constant to be chosen to determine the growth rate of depth. In this work, we demonstrate that the depth of BPNN in hybrid quantum-classical computing can be $O(1)$ and the upper bounds on the number of weight/gates are $O((1/\epsilon)^{d/n})$ (both quantum and classical computers). Apparently, our upper bounds are even approching to the lower bound of the networks on classical computers, which are unquantized and should have stronger expressive power. This clearly demonstrates the potential quantum advantage that can be attained.

Figure 3: Illustration of three basic neural computation used for the prologue-acceleration-epilogue computing scheme: (a) function $f_c$ to obtain constant $c$; (b) function $f_m$ for multiplication; (c) function $f_{\theta,k}$ with four ReLU functions in terms of the expansion point $k/S$ to act as a selector; (d) the output of function $f_{\theta,k}$ in terms of input $k$.

# 4 NEURAL NETWORK IN HYBRID QUANTUM-CLASSICAL COMPUTING SCHEME AND ITS UNIVERSAL APPROXIMABILITY

## 4.1 NEURAL NETWORK IN "PROLOGUE-ACCELERATION-EPILOGUE" SCHEME

A trivial solution for the hybrid quantum-classical computing scheme in Figure 2 is to do nothing on the quantum computer during the acceleration phase and load all the computations simply on the classical computer during the prologue or epilogue phases. In this case, all existing results on universal approximability and complexity bounds for classical computing can be readily applied. However, such a solution does not exploit any quantum power and thus is of little interest.

Accordingly, we add the constraint that when implementing a neural network, the computation in the quantum acceleration phase should be at least of the same asymptotic complexity compared with that in the classical prologue and epilogue phases.

With full consideration of the limitations and advantages of the quantum acceleration, we apply the most basic neuron operations: the binary weighted sum and the polynomial activation function. Such a network is called binary polynomial neural network (BPNN) in this paper. Let $\boldsymbol{x}$ be the $d$-dimensional input, $\boldsymbol{x} \in [0,1]^d$. We define the neuron operation in BPNN to be a function $O : \boldsymbol{x} \to y$, where $y \in [-1,1]$, which can be formulated follows:

$$O(\boldsymbol{x}) = \sigma\left(\boldsymbol{w}^T \boldsymbol{x} + b\right) \tag{1}$$

where $\boldsymbol{w} \in \{-1,+1\}^d$ represents a vector of binary weights; $b \in [0,1]$ is the bias; $\sigma$ is the activation function, which can be a polynomial function. Kindly note that bias $b$ can be relaxed to $b \in \mathbb{R}$ and $\sigma$ to polynomial or ReLU functions for the epilogue phase.

## 4.2 UNIVERSAL APPROXIMATION AND ERROR BOUND OF BPNN

The function space $\mathcal{F}_{d,n}$ considered in this work is defined as

$$\mathcal{F}_{d,n} = \{f \in \mathcal{W}^{n,\infty}([0,1]^d) : \max_{||\boldsymbol{n}||_1 \leq n} \operatorname{ess\,sup}_{x \in [0,1]^d} |D^{\boldsymbol{n}} f(\boldsymbol{x})| \leq 1.\} \tag{2}$$

where $\mathcal{W}^{n,\infty}([0,1]^d)$ is the Sobolev space on $[0,1]^d$ consisting of functions lying in $L^\infty$ along with their weak derivatives $D^{\boldsymbol{n}}$ up to order $n$. Kindly note that the weak derivative indicates that $f$ is not necessary to be differentiable and the function space $\mathcal{F}_{d,n}$ includes a wide class of functions.

In this subsection, we are going to show that for any target function $f \in \mathcal{F}_{d,n}$, there is a function $f_2$ with a particular form that can approximate $f$ with arbitrarily small error. This particular form of $f_2$ will enable us to realize it with BPNN and implement it with our hybrid quantum-classical computing scheme precisely and efficiently, which gives the universal approximability.

We start with the construction of two basic functions in BPNN: (1) obtaining an arbitrary constant; (2) conducting multiplications. To obtain an arbitrary constant within the range $[0,1]$, we formulate a one-layer neuron as follows.

**Proposition 4.1.** *Let $f_c$ be a sub-network of BPNN with only two weight values -1 and +1. An arbitrary constant $c$ can be obtained by $f_c$, such that the approximation error $\epsilon_c = 0$.*

By setting $\boldsymbol{w} = (+1,-1)^T$, $\boldsymbol{x} = (x,x)^T$, and $b = c$, an arbitrary constant $c$ can be obtained, as shown in Figure 3(a). Once the constant is obtained, we can get Proposition 4.2 to carry out

multiplication between a constant $c$ and a variable $x$. It can also obtain the multiplication of a pair of variables $x$ and $y$.

**Proposition 4.2.** *Let $f_m$ be a sub-network of BPNN with only two weight values -1 and +1. Given variable $x$ and variable $y$ (or constant $y$), the multiplication $xy$ can be derived from $f_m$, such that the approximation error $\epsilon_m = 0$.*

Since the square function is provided in $\sigma$, we can obtain $4xy$ based on the fact that $(x+y)^2 - (x-y)^2 = 4xy$. According to Proposition 4.1, we can create a constant scaling factor to adjust the value from $4xy$ to $xy$. Details please see Figure 3(b) and refer to Appendix A.1.

We employ a function $\psi_{\boldsymbol{k}}(\boldsymbol{x})$ to perform a "selection" operation. Considering the input has one variable, we divide $f_t$ on segment $[0, 1]$ to $S$ segments, which provides $S$ points for the Taylor expansion. At each point, $k \in [0, S]$, it is corresponds to one Taylor expansion, denoted as $f_t^k$, as shown in Figure 2. At run time, all these functions take the inputs for execution, and at the end of the neural network, they go through a "selection function" to extract the nearest expansion point in terms of inputs. For instance, if expansion point $x = 0.25$ and the step of $S$ segments is $0.1$, only $f_t^2$ and $f_t^3$ may contribute to the final result. In our implementation, for $x$ around the expansion point $\frac{k}{S}$ (i.e., $\frac{3k-2}{3S} \leq x \leq \frac{3k+2}{3S}$), it can be approximated using $f_t^k$ (i.e., by multiplying 1); however, it cannot be approximated by $f_t^m$ where $m < \frac{3k-2}{3S}$ or $m > \frac{3k-2}{3S}$ (i.e., by multiplying 0). To enable the above function, we apply the basic neuron operation to implement function $h(x, \frac{k}{S})$.

$$h(x, \frac{k}{S}) = \begin{cases} 1 & |x - \frac{k}{S}| \leq \frac{1}{3S} \\ 2 - 3S \cdot |x - \frac{k}{S}| & \frac{1}{3S} < |x - \frac{k}{S}| < \frac{2}{3S} \\ 0 & otherwise \end{cases} \tag{3}$$

Extending to the case of $d > 1$, for $\boldsymbol{k} = (k_1, \cdots, k_d) \in \{0, 1, \cdots, S\}^d$, we have $\psi_{\boldsymbol{k}}(\boldsymbol{x})$ defined as

$$\psi_{\boldsymbol{k}}(\boldsymbol{x}) = \prod_{i=1}^d h(x_i, \frac{k_i}{S}), \tag{4}$$

**Proposition 4.3.** *Let $f_\theta$ be a sub-network of BPNN with only two weight values $-1$ and $+1$. Given the expansion point $\frac{\boldsymbol{k}}{S}$, the function $\psi_{\boldsymbol{k}}(\boldsymbol{x})$ can be implemented by $f_\theta$ using the ReLU function and the multiplication implemented by $f_m$.*

As shown in Figure 3(c), we apply four ReLU functions to implement function $\frac{h(x, \frac{k}{S})}{3S}$. Basically, we apply $ReLU(x - y)$, where $y \in \{a, b, c, d\}$. Specifically, for $ReLU(x - a)$, it creates the first turning point, then at point $b$, it uses $ReLU(x - a) - (ReLU(x - b))$ to create the second turning point. Finally, the figure in Figure 3(c) can be obtained. Then, we scale it up to $h(x, \frac{k}{S})$. Kindly note that segments $[a, b]$ and $[c, d]$ in Figure 3(c) will be overlapped to the expansions at $\frac{k-1}{S}$ and $\frac{k+1}{S}$. Lastly, by multiplying $h(x, \frac{k}{S})$ we can obtain $\psi_{\boldsymbol{k}}(\boldsymbol{x})$ in the case of $d > 1$.

After showing that BPNN can realize the above operations, we are ready to demonstrate the universal approximation property of BPNN by proving that it can implement a complex function that approximates any $f \in \mathcal{F}_{d,n}$ with arbitrarily small error.

**Lemma 4.4.** *For any $f \in \mathcal{F}_{d,n}$, there exists a function $f_2 = \sum_{\boldsymbol{k} \in \{0, \cdots, S\}^d} \psi_{\boldsymbol{k}} \sum_{||\boldsymbol{v}||_\infty < n} c_{\boldsymbol{k}, \boldsymbol{v}} \boldsymbol{x}^{\boldsymbol{v}}$ that can be realized by a BPNN and can approximate $f$ with error $\delta \leq \frac{2^d}{n!} \left(\frac{d}{S}\right)^n$ where $S$ and $c_{\boldsymbol{k}, \boldsymbol{v}}$ are constant, $\boldsymbol{v} \in \{0, 1, \cdots, n-1\}^d$.*

The proof utilizes the function approximation idea from Yarotsky (2017). We partition the unity on $[0, 1]^d$ and approximate $f$ using the Taylor polynomial of order $n-1$, denoted as $f_t$. Then, we prove that the approximation function can be rewritten to the given form and the approximation error can be bounded to an arbitrarily small $\delta$. The complete proof is in Appendix A.2.

In summary, we have shown that BPNN can realize a few basic functions through the propositions. Then in Lemma 4.4, we show that by using these simple functions as the building block, we are able to approximate any $f$ by a function $f_2$ realized by a BPNN with arbitrarily small error, which gives the universal approximation property and error bound. The expansion point at $k/S$ in Figure 2 gives an example to obtain $f_2$ when $\boldsymbol{k} = k$ and $d = 1$. In the figure, $f_{\theta,0}$ to $f_{\theta,S}$ represents $\psi_{\boldsymbol{k}}$. For $\boldsymbol{k} = k$, $f_k(x)$ in the figure represents $\sum_{||\boldsymbol{v}||_\infty < n} c_{k, \boldsymbol{v}} \boldsymbol{x}^{\boldsymbol{v}}$.

### 4.3 APPROXIMATION PROCEDURE IN "PROLOGUE-ACCELERATION-EPILOGUE"

In order to construct a BPNN network in the hybrid quantum-classical scheme to obtain the bounds, we need to solve the following problem: Given a function $f$ in function space $\mathcal{F}_{d,n}$, an error bound $\epsilon > 0$, and a set of inputs, the neural network accelerated by the quantum computer and finalized in the epilogue needs to approximate the given function $f$, such that the approximation error is no more than $\epsilon$. Based on Lemma 4.4, we take Taylor polynomial as a bridge and construct BPNN on the hybrid quantum-classical computer to approximate the Taylor polynomial to establish the bounds. Specifically, we need to determine whether to use quantum computer or classical computer to implement each function at the expansion point $k$, i.e., $f_t^k$. As illustrated in Figure 2, the prologue phase conducts data preparation, the acceleration phase accelerates Taylor expansion at all points, and the epilogue phase implements the selection function. In the following texts, we use the expansion point at $\frac{k}{S}$ as an example to demonstrate how the hybrid computing scheme works.

**Prologue phase.** The classical computer conducts the data preparation: it encodes $n$ input data (including the variable and its coefficient) into $\log n$ qubits. We apply the same data encoding method in (Bravo-Prieto et al., 2020; Jiang et al., 2020), that is, constructing an unitary matrix $A$, such that all inputs are normalized to the first column vector $A_1$ in $A$. Then $A_1$ is encoded to the quantum states. Limited by the data representation of qubits, we have $||A_1|| \leq 1$. If $||A_1|| < 1$, we can add an additional dummy value to make sure the sum of inputs to be 1; while if $||A_1|| > 1$, we scale all the inputs to make sure that they can be encoded to $\log n$ qubits. As pointed by Bravo-Prieto et al. (2020), a single column in a unitary matrix $A$ can be decomposed to the quantum circuit with gate complexity of $O(\log n)$, where $\log n$ is the number of qubits. In our case, we only need to decompose the column $A_1$ in the matrix $A$ for quantum-state preparation on $\log n$ qubits.

**Quantum acceleration phase.** The function of Taylor polynomial is implemented on the quantum computer. Compare to the classical computer, quantum computing has limitations that restrict the operations in the neural networks.

- First, non-linear functions such as ReLU needs to be implemented as classical Boolean circuit with duplicate registers, which incurs high cost.
- Second, since the computation is based on the amplitude, it has the constraint that the real part of all data should range from -1 to 1.

For the non-linearity issue, it can implement the quadratic or even higher-order polynomial function by repeatedly executing the same operations on different qubits. For the data range issue, we can scale the inputs and outputs on the classical end. In addition, the quantum computer has obviously advantages over the classical counterpart. It can use $\log n$ qubits to represent $n$ data and achieve massive parallelism. Section 4.4 will present a design to take full use of such an advantage.

**Epilogue phase.** After the computation intensive tasks completed by the quantum acceleration, the epilogue phase collects data for all expansion points, and selects the correct one in terms of the input to formulate the final result. We move the selection procedure to classical computer to take advantage of the low-cost ReLU non-linear function. Specifically, we apply 4 ReLU functions to formulate a function shown in Figure 3(c). After this, we sum up all the results. Since the selection function will prune the results if the inputs do not belong to the expansion points, the output vector has large sparsity with at most $2^d$ (i.e., 2 for $d = 1$) non-zero values, among over $d^n S$ outputs. Therefore, it can be performed efficiently on the classical computer.

### 4.4 IMPLEMENTATION OF NETWORKS IN QUANTUM ACCELERATION

The quantum acceleration phase implements a set of sub-networks to obtain the output of Taylor polynomial at all expansion points, i.e., $\forall k \in \{0, \cdots, S\}^d$, $f_t^k$. Since the structure of all sub-networks are the same, we focus on one sub-network $net_k$ to implement function $f_t^k$.

Before introducing the detailed implementation, we first define a quantum sub-system $Q_i^m$ to be the $i^{th}$ sub-system which is composed of $m$ qbits to represent $2^m$ inputs at most. Then, we define notation "$\otimes$" between $Q_i^{m1}$ and $Q_j^{m2}$ to be the tensor product of these two quantum sub-systems. For example, for $m1 = 2$ and $m2 = 2$, we have $Q_1^{m1}$ to be a sub-system with two qbits $|\phi_0\phi_1\rangle$ and $Q_2^{m2}$ to contain $|\phi_2\phi_3\rangle$. Then $Q_1^{m1} \otimes Q_2^{m2} = |\phi_0\phi_1\rangle \otimes |\phi_2\phi_3\rangle = |\phi_0\phi_1\phi_2\phi_3\rangle$. Let $A\{|\phi_0\phi_1\rangle = |00\rangle\} = a_1$ be the amplitude of sub-system $Q_1^{m1}$ to be $|00\rangle$ state and $A\{|\phi_2\phi_3\rangle = |00\rangle\} = a_2$ for $Q_2^{m2}$. By combining these two sub-systems, we have system $Q_{1,2}^4$, where the amplitude for state

Figure 4: Illustration of quantum implementation of $f_t^k$: (a) implementing a set of parallel neuron computations with only one $H$ gate; (b) the multiplication function to achieve $c_0 x$; (c) the corresponding quantum circuit for $c_0 x$, with the state transitions represented in the rectangles.

$|0000\rangle$ is $A\{|\phi_0\phi_1\phi_2\phi_3\rangle = |0000\rangle\} = a_1 \times a_2$. Kindly note that such a combination is the base for the polynomial non-linear functions, since it can automatically multiply the corresponding states.

**Basic operations.** We now introduce the implementation of basic operations in BPNN on quantum computers, including (1) linear function $\boldsymbol{w}^T\boldsymbol{x}$, and (2) high-order polynomial activation function $\sigma$.

- For a *linear function $\boldsymbol{w}^T\boldsymbol{x}$*, it can be implemented in two steps: (1) encoding $d$ inputs in vector $\boldsymbol{x}$ to a system with $2^{\lceil \log(d+1) \rceil}$ states represented by $\lceil \log(d+1) \rceil$ qbits, where "1" is a dummy input to guarantee the sum of squared states to be 1; (2) encoding binary weights $\boldsymbol{w}$ to these $\lceil \log(d+1) \rceil$ qbits using Pauli-Z gates or Controlled-Z gates. For the linear function $\boldsymbol{w}^T\boldsymbol{x} + b$, the bias $b$ can be encoded along with the inputs, and add an additional weight with a value of $+1$.

- Let $y = \boldsymbol{w}^T\boldsymbol{x} + b$, the *quadratic function*, $\sigma(y) = y^2$, can be implemented by using two quantum sub-systems (e.g., $Q_1$ and $Q_2$) and each of which implements the same function to get $y$. At the end of these operations, both zero states $|0\cdots0\rangle$ in $Q_1$ and $Q_2$ are $y$, and the $|0\cdots0\rangle$ state in the combination of these two systems, $Q_{1,2}$, will be $y^2$. Similar to the implementation of $y^2$, the high-order non-linearity (e.g., $y^m$), can be implemented with $m$ quantum sub-systems.

Based on the above designs, the neuron operations can be implemented on the quantum circuit. We next consider the implementation of $f_t^k$ on top of these basic designs.

**High-parallelism and low-cost design.** An arbitrary design may not make full use of the high-parallelism of quantum computing and result in a high cost in circuit depth and width. We fully exploit the parallelism of quantum operations by using the following property of BPNN.

**Property 4.5.** *For an $n$-input neuron in BPNN whose activation function is polynomial linearity or non-linearity, it can be decomposed to $\lceil \log_2 n \rceil$ layers, such that each neuron has at most 2 inputs.*

A network construction can be carried out to complete the transformation. We can divide $n$ inputs to $\lceil \frac{n}{2} \rceil$ groups, each of which contains 2 inputs, and then apply their corresponding weights in the first layer and use $\{+1, +1\}$ for all the remaining layers. The linear function is applied as the activation for the intermediate layers, and the original activation is applied to the last layer. For the sub-network $net_k$ for function $f_t^k$, the network itself has this property in calculating all terms in Taylor polynomial (see Figure 2). In addition, all the weights for these neuron operations are either $\{+1, +1\}$ or $\{+1, -1\}$ (see Figure 3(b)). This allows us to take advantage of massive parallelism provided by quantum computing to accelerate these operations.

**Proposition 4.6.** *Let $net$ be a layer in BPNN with 2-input neurons in total, and there are $m$ neurons in total. Let $Q^k$ be a quantum system with $k$ qubits, and $2m$ inputs of $net$ are encoded to $2^k$ states in $Q^k$. If all neurons have the same weights, then all $m$ neuron computations can be completed in 3 steps with at most 3 basic quantum logic gates.*

The proof and the details of the quantum circuit constructed are included in Appendix A.3. In general, this proposition indicates that quantum computer can significantly accelerate a batch of neuron computations with extremely low cost. Take a further step, we have Proposition 4.7.

**Proposition 4.7.** *Let $net$ be a layer in BPNN with 2-input neurons in total, and there are $2 \times m$ neurons in total, and each 2-adjacent (pair) neurons share the same inputs. Let $Q^k$ be a quantum system with $k$ qubits, and $2m$ inputs of $net$ are encoded to $2^k$ states in $Q^k$. If each pair of neurons has odd number of $+1$ weight and all pairs of neuron have the same weights, then all $2m$ neuron computations can be completed in 3 steps with at most 3 basic quantum logic gates.*

The proof and the details of the above proposition is included in Appendix A.4. We can see that the neuron operations in sub-network $net_k$ for function $f_t^k$ satisfy the above condition, where each pair

of neurons with the shared inputs has odd number (i.e., 3) +1 weights, and all pairs of neurons have the same weights. In fact, for this specific weights, all these neuron computations can be conducted in parallel with only 1 Hadamard $H$ Gate, as shown in Figure 4(a).

**Quantum design of sub-network for multiplication function $f_m$.** Multiplication is the core operation in the sub-network $net_k$ for function $f_t^k$. In particular, each term of Taylor polynomial is only composed of the multiplication between "constant and variable" or "variables". We demonstrate the multiplication operation in Figure 4(b)-(c). For the simplicity of presentation, we assume that the inputs $x$ and $c_0$ can be encoded into 2 states (i.e, $x^2 + c_0^2 = 1$) in two quantum sub-systems ($Q_1^1$ and $Q_2^1$), otherwise, we can add dummy states using an additional qbit for the encoding. In this example, we employ two sub-systems $Q_1^1$ and $Q_2^1$ for the quadratic function, and a $CNOT$ gate is applied to adjust the position to make the square terms in the front of all states. Then, two $H$ gates are applied for the second-layer neuron computation. Finally, the multi-controlled not gates are applied to extract the amplitude (i.e., $c_0 \times x$) to an Ancilla qbits for measurement.

Kindly note that since the measurement is on the probabilistic domain while the computation is based on the amplitudes, a square operation will be automatically conducted. To make the whole system consistent, we use the square root on each input and coefficient during the encoding. Another observation is that every term in Taylor polynomial has one state, if we extract all terms at the end of the procedure, it will lead to high cost. To overcome this, we observe that we can accumulate the results using $H$ gates with a scale of $\frac{1}{\sqrt{2}}$ for each $H$ gate. As a result, we only need one multi-controlled NOT gate for each quantum sub-systems.

**Proposition 4.8.** *If the order of a term be $k$, its corresponding sub-network in BPNN contains $\lceil \log k \rceil$ layers. Then, the implementation on a quantum computer involves $2k$ quantum sub-systems.*

Without loss of generality, we consider $k = 2^m$ and $m$ is a positive integer (otherwise, we can add dummy 1s). We can apply the divide-and-conquer approach to compute a pair of variables at each time. As a result, the operation can be completed in $m$ layers. For each multiplication, we need 2 quantum sub-systems as shown in Figure 4(c). Therefore, it involves $2k$ quantum sub-systems.

Based on the above design, any terms in Taylor polynomial can be implemented. Kindly note that the real number is applied in each step, and therefore, there will be no computation error for the implementation of $f_t^k$ on quantum computer.

## 5 CONCLUSION AND INSIGHTS

Although the implementation of neural network in quantum computing is still in its infancy, results in this work provide theoretical and practical insights for the design of neural networks for quantum computing to fully harvest the quantum power in the hybrid quantum-classical computing scheme.

- Neural networks designed for hybrid quantum-classical computing, including TensorFlow Quantum, have the ability to approximate a wide class of functions with arbitrarily small error.
- For the near-term hybrid quantum-classical neural network designs, the proposed prologue-acceleration-epilogue architecture is a promising computing scheme to achieve high accuracy with only two interfaces between quantum and classical portions for data conversion.
- Based on the depth complexity of $O(1)$ for a hybrid quantum-classical computing scheme, it inspires us that the design of network may consider a "shallow" quantum circuit, instead of the "deep" version on classical computer. The power of shallow circuits has been demonstrated in the recent works (Benedetti et al., 2019; Cerezo et al., 2020).

Putting all together, this work demonstrated the combination of machine learning and quantum computing is a promising research direction, and the results can guide future research works in the design of neural networks for quantum computing.

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

# A   PROOFS

## A.1   PROOF OF PROPOSITION 4.2

**Proposition 4.2.** *Let $f_m$ be a sub-network of BPNN with only two weight values -1 and +1. Given variable $x$ and variable $y$ (or constant $y$), the multiplication $xy$ can be derived from $f_m$, such that the approximation error $\epsilon_m = 0$.*

*Proof.* We show the correctness of the proposition by constructing a 2-layer neuron network using the basic neuron operations for both multiplications between "variable and constant" and "variables".

First, we construct the multiplication of $c \times x$ between a variable $x$ and a constant $c$ (i.e., $y$). As shown in Figure 3(b), we first using Proposition 4.1 to form a constant $\frac{c}{4}$. Let $\boldsymbol{x} = (x, \frac{c}{4})^T$ be a 2-dimension input and $z$ be the output; and let $\boldsymbol{w}_{i,j}$, $b_{i,j}$ and $\sigma_{i,j}$ be the weights, bias, and activation function of the $j^{th}$ output neuron at the $i^{th}$ layer, respectively. We set $\boldsymbol{w_{1,1}} = \{+1, +1\}$, $\boldsymbol{w_{1,2}} = \boldsymbol{w_{2,1}} = \{+1, -1\}$, $b_{1,1} = b_{1,2} = b_{2,1} = 0$, $\sigma_{1,1}$ and $\sigma_{1,2}$ be the quadratic function, and $\sigma_{2,1}$ be the linear function. Then, we can represent $c \times x$ using two layers of the neuron operations: $o_{1,j} = \alpha_{1,j} \times \sigma_{1,j} \times (\boldsymbol{w_{1,j}^T x} + \boldsymbol{b_{1,j}})$ and $z = \alpha_{2,1} \times \sigma_{2,1} \times (\boldsymbol{w_{2,1}^T o} + \boldsymbol{b_{2,1}})$, where $\boldsymbol{o} = (o_{1,1}, o_{1,2})^T$.

Second, the multiplication of $xy$ between two variables can be constructed in a similar way. The only difference is that we need an additional multiplication to scale the results of $4xy$ to $xy$. Specifically, we can initialize two inputs as $4xy$ and $\frac{1}{16}$.  $\square$

## A.2   PROOF OF LEMMA 4.4

**Lemma 4.4.** *For any $f \in \mathcal{F}_{d,n}$, there exists a function $f_2 = \sum_{\boldsymbol{k} \in \{0,\ldots,S\}^d} \psi_{\boldsymbol{k}} \sum_{||\boldsymbol{v}||_\infty < n} c_{\boldsymbol{k},\boldsymbol{v}} \boldsymbol{x^v}$ that can be realized by a BPNN and can approximate $f$ with error $\delta \leq \frac{2^d}{n!} \left(\frac{d}{S}\right)^n$ where $S$ and $c_{\boldsymbol{k},\boldsymbol{v}}$ are constant, $\boldsymbol{v} \in \{0, 1, \ldots, n-1\}^d$.*

*Proof.* We define the approximation to $f$ as

$$f_2 \triangleq \sum_{\boldsymbol{k} \in \{0,\ldots,S\}^d} \psi_{\boldsymbol{k}} P_{\boldsymbol{k}}. \tag{5}$$

where $P_{\boldsymbol{m}}(\boldsymbol{x})$ is the order $n - 1$ Taylor polynomial at $\frac{\boldsymbol{k}}{S}$. Specifically, we have

$$P_{\boldsymbol{k}}(\boldsymbol{x}) = \sum_{||\boldsymbol{n}||_1 < n} \frac{D^{\boldsymbol{n}} f}{\boldsymbol{n}!} \Big|_{\boldsymbol{x} = \frac{\boldsymbol{k}}{S}} \left(\boldsymbol{x} - \frac{\boldsymbol{k}}{S}\right)^n. \tag{6}$$

where $\boldsymbol{n} = (n_1, \ldots, n_d) \in \{0, 1, \ldots, n-1\}^d$. The approximation error to $f$ can bounded by Equation (7).

$$\begin{aligned} |f(\boldsymbol{x}) - f_2(\boldsymbol{x})| &= |\sum_{\boldsymbol{k}} \psi_{\boldsymbol{k}}(\boldsymbol{x})(f(\boldsymbol{x}) - P_{\boldsymbol{k}}(\boldsymbol{x}))| \\ &\leq 2^d \max_{\boldsymbol{k}:|x_i - \frac{k_i}{S}| < \frac{1}{S} \forall i} |f(\boldsymbol{x}) - P_{\boldsymbol{k}}(\boldsymbol{x})| \\ &\leq \frac{2^d d^n}{n!} \left(\frac{1}{S}\right)^n \max_{|\boldsymbol{n}|=n} \operatorname{ess\,sup}_{\boldsymbol{x} \in [0,1]^d} |D^{\boldsymbol{n}} f(\boldsymbol{x})| \\ &\leq \frac{2^d}{n!} \left(\frac{d}{S}\right)^n \end{aligned} \tag{7}$$

The first line follows $\sum_{\boldsymbol{k} \in \{0,\ldots,S\}^d} \psi_{\boldsymbol{k}}(x) = 1$. The second step is based on the fact that any $x \in [0, 1]^d$ is on the support of at most $2^d$ $\psi_{\boldsymbol{k}}$ functions. In the third step, we use the Lagrange's form of the Taylor reminder and lastly we use $\max_{\boldsymbol{n}:|\boldsymbol{n}|=n} \operatorname{ess\,sup}_{\boldsymbol{x} \in [0,1]^d} |D^{\boldsymbol{n}} f(\boldsymbol{x})| \leq 1$.

Using the binomial theorem, $f_2$ can be rewritten as

$$f_2 = \sum_{\boldsymbol{k} \in \{0,\ldots,S\}^d} \psi_{\boldsymbol{k}} \sum_{||\boldsymbol{n}||_1 < n} \sum_{\boldsymbol{v}:\boldsymbol{v} \leq \boldsymbol{n}} c_{\boldsymbol{k},\boldsymbol{n},\boldsymbol{v}} \boldsymbol{x}^{\boldsymbol{v}}$$
$$= \sum_{\boldsymbol{k} \in \{0,\ldots,S\}^d} \psi_{\boldsymbol{k}} \sum_{||\boldsymbol{v}||_\infty < n} c_{\boldsymbol{k},\boldsymbol{v}} \boldsymbol{x}^{\boldsymbol{v}} \tag{8}$$

where $\boldsymbol{v} = (v_1,\ldots,v_d) \in \{0,1,\ldots,n-1\}^d$ and $\boldsymbol{v} \leq \boldsymbol{n}$ means $v_i \leq n_i$ for $i \in \{1,\ldots,d\}$. Kindly note that the number of terms $c_{\boldsymbol{k},\boldsymbol{v}} \boldsymbol{x}^{\boldsymbol{v}}$ in $f_2$ is $|\boldsymbol{k}| \cdot |\boldsymbol{v}| \leq (S+1)^d \sum_{i=0}^{n-1} d^i = (S+1)^d (\frac{d^n-1}{d-1})$. For the implementation of each term in BPNN, it involves the multiplication of a constant and an input variable with the $n^{th}$ order non-linear function, which can be implemented using the basic neural operations in Figure 3 with a constant number of weights (i.e., 2 for constant generation and 6 for multiplication). Then, all the terms will be sum up using a linear activation function, and the number of weights is the same as the number of terms. As a result, the complexity of the corresponding neural network is $O(S^d d^{n-1})$.

Given Proposition 4.1, 4.2, and 4.3, we can conclude that $f_2$ can be realized by a BPNN. □

## A.3 PROOF OF PROPOSITION 4.6

**Proposition 4.6.** *Let net be a layer in BPNN with 2-input neurons in total, and there are $m$ neurons in total. Let $Q^k$ be a quantum system with $k$ qbits, and $2 \times m$ inputs of net are encoded to $2^k$ states in $Q^k$. If all neurons have the same weights, then all $m$ neuron computations can be completed in 3 steps with at most 3 basic quantum logic gates.*

*Proof.* First, the 2-input neurons have possible 4 pairs of weights: $\{(+1,+1),(+1,-1),(-1,+1),(-1,-1)\}$. Let's first consider $Q^1$, where we have $I = (x_1,x_2)^T$.which is introduced in the following texts Then, the gate applied on the qbit will conduct the matrix and vector multiplication. Therefore, we can regard each row in the matrix representation of gates to be the weight pairs.

First, we know that the matrix representation of $H$ is $\frac{1}{\sqrt{2}} \begin{bmatrix} 1 & 1 \\ 1 & -1 \end{bmatrix}$, it contains two weights pairs $(+1,+1)$ and $(+1,-1)$.

Second, we can use $Z \times X \times H$ to get $Z \times X \times H = \frac{1}{\sqrt{2}} \begin{bmatrix} 1 & 0 \\ 0 & -1 \end{bmatrix} \times \begin{bmatrix} 0 & 1 \\ 1 & 0 \end{bmatrix} \times \begin{bmatrix} 1 & 1 \\ 1 & -1 \end{bmatrix} = \begin{bmatrix} 1 & -1 \\ -1 & -1 \end{bmatrix}$ Now, we have the weight pair $(-1,-1)$.

Third, we can apply $H \times X$ to get matrix which contains weight pair $(-1,1)$. We have $H \times X = \frac{1}{\sqrt{2}} \begin{bmatrix} 1 & 1 \\ 1 & -1 \end{bmatrix} \times \begin{bmatrix} 0 & 1 \\ 1 & 0 \end{bmatrix} = \begin{bmatrix} 1 & 1 \\ -1 & 1 \end{bmatrix}$.

The above results demonstrate that at most 3 quantum gates (the second case) are needed to cover all weight pairs.

Next, we demonstrate that no additional gates are needed for computing multiple neurons with the same weights. To demonstrate this, let's first consider the tensor product operator, which is applied to obtain the matrix of parallel gate. For instance, $H \otimes I = \frac{1}{\sqrt{2}} \begin{bmatrix} 1 & 0 \\ 0 & 1 \end{bmatrix} \otimes \begin{bmatrix} 1 & 1 \\ 1 & -1 \end{bmatrix} = \frac{1}{\sqrt{2}} \begin{bmatrix} 1 & 1 & 0 & 0 \\ 1 & -1 & 0 & 0 \\ 0 & 0 & 1 & 1 \\ 0 & 0 & 1 & -1 \end{bmatrix}$. The observation here is that by tensor product of the gate and identity gate $I$, the weights for each pair of inputs are the same. Kindly note that identity gate $I$ indicates that no gate is placed and no gate cost at all.

Therefore, we can conduct $2 \times m$ operations at the same time with at most 3 gates to cover the weight pair. □

A.4   PROOF OF PROPOSITION 4.7

**Proposition 4.7.** *Let net be a layer in BPNN with 2-input neurons in total, and there are $2 \times m$ neurons in total, and each 2-adjacent (pair) neurons share the same inputs. Let $Q^k$ be a quantum system with $k$ qubits, and $2 \times m$ inputs of net are encoded to $2^k$ states in $Q^k$. If each pair of neurons has odd number of $+1$ weight and all pairs of neuron have the same weights, then all $2 \times m$ neuron computations can be completed in 3 steps with at most 3 basic quantum logic gates.*

The proof of this proposition is similar to Proposition 4.6. From the proof procedure in Proposition 4.6, it is clear to see that one gate can perform 2 neuron computation at the same time, meanwhile the values of weights are different. In addition, due to the matrix representation of a gate need to be unitary, we cannot engagement the same weights in one gate operation. It is equivalent to that the number of both weight $+1$ and weight $-1$ needs to be an odd number.

A.5   PROOF OF RESULT 3.1

**Results 3.1.** *For any given function $f \in \mathcal{F}_{d,n}$ and an expansion point $k$, its Taylor polynomial at point $k$ can be implemented on the quantum computer, such that $(i)$ the network can exactly implements the Taylor polynomial, $(ii)$ the depth is $O(\log n)$, $(iii)$ the number of gates is $O(n^2 \log n \log d)$, $(iv)$ the number of qbits is $O(n^2 \log d)$.*

*Proof.* The cost of quantum circuit comes from 4 parts. Figure 4(c) demonstrates the first 3 parts: computation (i.e., $H$ gate), state position adjustment (i.e., $CNOT$ gate), and state extraction (i.e., multi-controlled NOT gate). In addition, as stated, we apply $H$ gate in each sub-system to sum up results of the $i^{th}$ order term.

In our design, the gate complexity for $H$ is $O(n \log n)$, since each layer of each sub-system needs 1 $H$ gate, and there are $\log n$ layers (see Proposition 4.8) and $n$ sub-systems.

For $CNOT$, for each layer at each sub-system, the number of $CNOT$ is half of the number of qbits, which is $(n + 1) \log(d)$. There are $\log n$ layers and $n$ sub-systems, and therefore, the complexity is $(n^2 \log n \log d)$.

Next, for the $H$ gates used for summation, the usage is related to the number of terms to be summed up. For the $i^{th}$ order term, it involves $d^i$ addition, while we can use $H$ gate as a adder tree, which will involve $\log(d^i) = i \log d$ gates. Therefore, the upper bound is $O(n \log n)$.

Finally, for multi-controlled NOT gate, each sub-system needs one multi-controlled NOT gate, where the number of control ports is the same as the number of qbits. A k-controlled NOT gate can be broken down to $poly(k)$ standard Toffoli gates. Therefore, for $n$ sub-systems and each sub-system with $O(n \log d)$ qbits, the cost complexity for the multi-controlled NOT gate is $O(n^2 \log d)$.

Overall, the cost bound of the quantum implementation for $f_t^k$ is $O(n^2 \log n \log d)$.   □

A.6   PROOF OF RESULT 3.2

**Results 3.2.** *For any given function $f \in \mathcal{F}_{d,n}$, there is a binary polynomial neural network with a fixed structure that can be implemented in the hybrid quantum-classical computing scheme, such that $(i)$ the network can approximate $f$ with any error $\epsilon \in (0,1)$, $(ii)$ the overall depth is $O(1)$; $(iii)$ the number of quantum gates is $O\left((1/\epsilon)^{\frac{d}{n}}\right)$; $(iv)$ the number of qbits is $O\left((1/\epsilon)^{\frac{d}{n}}\right)$; $(v)$ the number of weights on classical computer is $O\left((1/\epsilon)^{\frac{d}{n}}\right)$.*

*Proof.* In Lemma 4.4, we have already proved that a BPNN $f_2$ can approximate the given function $f$, whose approximation error $|f - f_2| = \delta \leq \frac{2^d}{n!}\left(\frac{d}{S}\right)^n$. To satisfy the given error $\epsilon$, we have:

$$\frac{2^d}{n}\left(\frac{d}{S}\right)^n \leq \epsilon \tag{9}$$

$$S \geq d \times \left(\frac{2^d}{n}\right) \times \left(\frac{1}{\epsilon}\right)^{\frac{1}{n}} \tag{10}$$

On the other hand, according to Lemma 3.1, we have the gate number used for implementing Taylor polynomial at one expansion point is $O(n^2 \log n \log d)$, and the number of qbits to be $O(n^2 \log n)$. For the whole system, there are $d$ variables in total, and each of which is partitioned into $S$ segments. Therefore, there are $S^d$ possible expansion points for the Taylor polynomial. Overall, the quantum gate number used in approximating the given function $f$ is $O(n^2 \log n \log d \times S^d)$, which is $O\left((1/\epsilon)^{\frac{d}{n}}\right)$; similarly, the number of qbits used for approximation is $O(n^2 \log n \times S^d)$, which is $O\left((1/\epsilon)^{\frac{d}{n}}\right)$. For depth, from Lemma 3.1, we have the depth of $O(\log n)$. In the epilogue phase on classical computer, the depth is 2 (see Figure 3 (a)). Therefore, the depth of the whole neural network is $O(1)$ in terms of the error bound $\epsilon$.

Finally, the weight cost on classical computer is proportional to the number of expansion points, because it take the output at each expansion point where 4 ReLU is operated. Therefore, the weight cost on classical computer is $O(S^n)$, which is $O\left((1/\epsilon)^{\frac{d}{n}}\right)$.

$\square$

## B  EXPERIMENTAL SETUP

We use pairwise classifiers in MNIST for the test the accuracy of different neural networks on different platforms, including (1) ReLU network in classical computing in Figure 1(a), (2) tree tensor network in hybrid quantum-classical computing in Figure 1(b), and (3) the BPNN in the prologue-acceleration-epilogue computing scheme in Figure 1(c). The results of the tree tensor network are obtained from Huggins et al. (2019). The result of the ReLU network in classical computing is conducted by constructing a ReLU neural network, where the input resolution is $28 \times 28$ pixels, the hidden layer has 16 neurons, and the output layer has 2 neurons for the classification. Then, we train the network using the learning rate of 0.01 on the Adam optimizer, the epoch of 10, the batch size of 16, and the input data to be normalized to make the sum of the square to be 1. The BPNN has the same structure as the ReLU neural network, except that the activation function of the hidden layer is the quadratic function.

Now, we introduce how we test the BPNN on the prologue-acceleration-epilogue computing scheme. Let denote $C1$ for prologue, $C2$ for epilogue and $Q$ for acceleration portion. In $C1$, we conduct the data prepossessing for quantum state-preparation. Then in $Q$, we apply the neural computation and the corresponding quantum circuit design in (Francesco et al., 2019; Jiang et al., 2020). Finally, in $C2$, we translate the output from the quantum data to classical data and then perform a fully connected layer for the final classification.

Next, the training of BPNN is carried out on a classical computer. Since all operations on $Q$ have the equivalent neural computation on classical computing, it is possible for us to train the network on a classical computer, which has the same training setup as the ReLU network.

Finally, we conduct the training and testing procedure for each pair of two classifiers in MNIST and obtain the results demonstrated in Figure 1.

