# OpenReview forum: "On the Universal Approximability and Complexity Bounds of Deep Learning in Hybrid Quantum-Classical Computing"
_ICLR.cc/2021/Conference — Reject_

### Official Review · AnonReviewer3 · 2020-10-28
**On the Universal Approximability and Complexity Bounds of Deep Learning in Hybrid Quantum-Classical Computing**

**Rating:** 4
**Confidence:** 4

**Review:**

The problem studied in this work is of interest in the quantum machine learning community, as the power of small and noisy quantum computers for machine learning problems is far from being understood. Therefore, it is important to study the expressivity of quantum neural networks as function approximators. This work uses the model introduced by Tensorflow Quantum, where different neurons can be implemented on either quantum or classical computers.

However, it is unclear how this result applies to current topologies of QNN/variational circuits used in current literature. From my knowledge of quantum variational circuits, the architecture proposed is different. To my understanding, this work addresses specifically the model proposed for TensorFlow quantum, and this should probably be made more explicit, as it's somehow different from current literature, where quantum neural network architectures are the sole computational node, and no classical computation is performed classically (besides the optimization of the parameters of the variational circuit). The model described in this work is called the "prologue, acceleration, epilogue".

If I understood the work properly, the role of quantum computers is to evaluate on a quantum computer the "binary" part of a Binary Polynomial Neural Network (this is what the authors call the acceleration phase), after in a prologue part the data is loaded as initial quantum state with a log-depth circuit.Then, in the epilogue phase, a nonlinearity, like a a ReLU function is applied classically,

It is really interesting the comparison with the approximation function of neural networks in classical computers. Perhaps more (recent) literature review on quantum expressiveness of other variational circuits can be added.

I checked some of the proofs in the manuscript, and they are correct. The paper is nicely written, but perhaps might benefit some more clarity, especially in the proofs in the appendix.

Overall, the submission would have benefited from experiments, showing that a QNN built with the architecture proposed in this work can achieve high accuracy in classification/regression tasks. This can be either done on small quantum computers, or even simulated in GPUs or large classical computers. Also, it would have been beneficial to write more clearly section 3.3, perhaps with an example, on how the classical optimizer is meant to choose the parameters of this circuit in a machine learning problem, i.e. how this architecture is meant to be used in practice.

Other remarks are the following:
- Please use a consistent notation for multiplication. If my understanding is correct, In proposition 3.2 and the subsequent lines, you use the notation $x \times y$, $xy$, and $x \dot y$ to denote the same operation.

-  In section 4.1 it's not clear to me what this sentence means:
 At the end of these operations, both zero states $\ket{0}$ in $Q_1$ and $Q_2$ are y, and the $\ket{0..}$ state in the combination of these two systems, $Q_{1,2}$ will be $y^2$. How can the zero register be $y^2$?

-  What is the $p$ in the Discussion section, when discussing the result of Yarotsky?

- I think in some parts of the paper the authors use $n$ for the number of qubits, and then $\log n$.

- Figure 1 could be split into two figures (left and right), and the notation in the figure could be better explained as the figure is referenced many times in the paper.

- Is written in the section where the  prologue phase is described "As pointed by Bravo-Prietoet al. (2020), unitary matrix A can be decomposed to the quantum circuit with gate complexity ofO(logn), where logn is the number of qbits." This is only true if the matrix $A$ is of the kind specified before, i.e. it's just one single column.

- The fact that the depth of BPNN in hybrid quantum-classical computing can be of $O(1)$ is a strong result that perhaps should be compared more with the literature on the power of quantum shallow circuits or constant depth circuits.

Also, I think the work should conform to the widespread and standard notation of using qubits and not qbits.

 Some typos:
 - Proof Proposition 4.2. "the Of" should be "the of".
 - After lemma 3.3 "which is introduced in the following texts" -> which is introduced next (or in the following section).
 - "To take the advantages of high-parallel in quantum computing, we made an observation on the network structure of BPNN as described in the following Property." Might be improved. It might be changed into "high-parallelism" and rephrased the whole sentence.
 -Section 4.2
 "d input variables and f has weak derivative" should be derivativeS.
 - . This brings flexibility in implementing functions (e.g., ReLU), while at the same calls for interface for massive data transfer between quantum and classical computers. Perhaps you wanted to say "at the same time calls for fast interfaces"

---

> ### Author Response · Authors · 2020-11-14
> **Response to AnonReviewer3**
>
> Thanks for your encouragement for the value of the studied problem. We appreciate your valuable time to give us helpful comments and check the correctness of the proofs. In Revision-1, we have carefully considered your comments and taken actions to address them. The detailed actions are reported as follows.
>
> **C1. How can the results apply to the current existing topologies of QNN/variation circuits?**
>
> Yes, as you said it addresses models for TensorFlow quantum and your understanding of the workflow for the proposed computing scheme is correct. In *Sec 3 of Revision-1*, we have clarified that the employed prologue-acceleration-epilogue computing scheme is a special case of TensorFlow quantum. In consequence, since we have proved that the neural networks designed for the target computing scheme have universal approximability, then we can derive that the quantum neural network designed for TensorFlow quantum also has the universal approximability.
>
> **C2. Literature review on quantum expressiveness of variational circuits.**
>
> Thanks for the suggestion, and we have added the literature review of quantum expressiveness in *Sec 2.2 of Revision-1*.
>
> **C3. Experiments on QNN built on the target computing scheme.**
>
> First of all, we would like to emphasize that similar to most existing works on neural network complexity bounds, the proof-by-construction method we used in this paper is not necessarily the best way to build QNNs for real tasks. It simply shows that a quantum neural network does exist to approximate a wide class of functions with arbitrarily small error (the reason why what we have derived is only an upper bound in Big-O notation). Depending on the actual problem, there can be much better ways to construct/train the network.
>
> Following your suggestion, in *Sec 2.2 of Revision-1*, we have added the experimental results on a classification task using the pairwise classifiers in MNIST dataset, as shown in Fig. 1. Using the classical ReLU neural network as a baseline, results have demonstrated that the QNN run on the proposed prologue-acceleration-epilogue computing scheme can achieve competitive classification accuracy with baseline, where the average accuracy gap is less than 0.5%.
>
> **C4. Adding examples to show the proposed architecture to be used in practice.**
>
> In *Appendix B of Revision-1*, we have added an example to demonstrate how to apply the existing QNN to the proposed architecture. Specifically, let denote C1 for the prologue, Q for the acceleration, and C2 for the epilogue. To use this platform, the input data can be prepared at C1 phase. Then, the existing QNN like that proposed in (Francesco et al, 2019; Jiang et al, 2020) can be implemented on Q portion for the neural computation. Finally, a ReLU layer can be an add-on to the output of Q for the post-processing. Following this procedure, we obtain almost the same accuracy with ReLU network for the pairwise classifiers in MNIST dataset, as shown in Fig. 1. In the whole procedure, it only needs two interfaces for the classical-quantum data transfer. Together with the results, it demonstrates that the proposed architecture can be used in practice.
>
> **C5. The statement of computing $y^2$ which is finally stored in zero state is not clear.**
>
> In *Sec 4.4 of Revision-1*, we have added discussions on how we compute the nth-order non-linear function (n>1). Specifically, we apply two sets of qubits (called sub-system), and they will first conduct the neural computation independently. The results are stored in the zero state for each sub-system. Then, by combining these two sub-systems (say S1 and S2) into one system (say S), the amplitude of the zero state for S will be the multiplication of amplitudes of zero states of S1 and S2. This is because the combination represents the tensor product. Therefore, for the computation of $y^2$, its final results will be stored at zero state of system S.
>
> **C6. The complexity of encoding unitary data to qubits is only true if it's just one single column.**
>
> In *Sec 4.3 of Revision-1*, we have clarified that the proposed architecture is to encode the data in vector $A_1$, and only the vector will be encoded to the qubits, which is the same case as described in the comment.
>
> **C7. Adding the literature on the power of quantum shallow circuits.**
>
> Thanks for the suggestion. In Revision-1, we have added new references, and correspondingly in *Sec 5 of Revision-1*, we have discussed the insights obtained by the depth complexity and demonstrated that the power of the shallow quantum neural network in the recent works.
>
> **C8. Comments and suggestions on splitting Figure 1 into two figures, notation definition (i.e., p of Yarotsky), notation consistency (i.e., multiplications and number of qubits), and typos.**
>
> Thanks for pointing out all these details. In Revision-1, per your suggestions, we have fixed these typos, added the missing definitions, and resolved the inconsistent issue.

---

### Official Review · AnonReviewer4 · 2020-10-29
**Universal approximation theorem for parameterized quantum circuits**

**Rating:** 6
**Confidence:** 4

**Review:**

The paper considers the expressivity and approximation properties of machine learning models where a parameterized quantum circuit is used to 'accelerate' a classical neural network. The results consider a model with a date encoder, a quantum circuit, and then a classical feedforward neural net for post-processing. To make the results non-trivial only models with asymptotically similar classical and quantum complexity are considered.

Using a technique best on Taylor polynomial approximations, the paper finds that a large class of smooth functions can be approximated using O(log(1/\epsilon)^(n/d)) quantum gates, qubits,  and classical width. This is slightly (logarithmic factors of  1/\epsilon) smaller than the best known upper bounds with ReLU or unconstrained quadratic networks.

Pros:

The paper shows how some techniques used to obtain classical approximation theorems can be extended to quantum gates. The slight improvement from the addition of a quantum element raises the possibility of showing a stronger separation in the future.

Cons:
The results in their current form do not really show any benefit (from the point of view of approximation) for quantum neural networks, since the bounds are only logarithmicallly better than classical bounds.

The technique used seems to be to use quantum units to mimic subfunctions in a manner quite similar to classical units, and does not provide much insight into why we should expect a 'quantum improvement' here.

---

> ### Author Response · Authors · 2020-11-14
> **Response to AnonReviewer4**
>
> Thanks for your encouragement and your valuable time to give us helpful comments. In Revision-1, we have carefully considered your comments. The detailed response is listed as follows.
>
> **C1. Pros: The paper shows how some techniques used to obtain classical approximation theorems can be extended to quantum gates. The slight improvement from the addition of a quantum element raises the possibility of showing a stronger separation in the future.**
>
> Thanks for the comment. Yes, the proof of universal approximability is the main contribution of this work. We use the bound-by-construction approach for the derivation and we further analyze the complexity bounds for the whole system.
>
> **C2. Cons: The results in their current form do not really show any benefit (from the point of view of approximation) for quantum neural networks, since the bounds are only logarithmically better than classical bounds.**
>
> The primary goal of this paper is to prove the universal approximability of neural networks on quantum computers, a missing piece in the literature. This question arises from the fact that the quantum circuit has limitations in computation operations to achieve high parallelism, such that classical neural networks cannot be directly mapped to quantum circuits. For example, quantum circuits do not allow common activation functions, like ReLU and Sigmod. This question is important since the failure to attain universal approximability will fundamentally hinder quantum neural networks from achieving state-of-the-art performance on various machine learning tasks, say image classification. And we prove it by using a bound-by-construction approach.
>
> Based on the constructed neural network, we further give the complexity bound analysis. Our conclusion is that neural networks on quantum computers can achieve a lower depth upper bound and smaller circuit size compared with those on classical computers. But it is in big O notation, so the actual reduction in depth and circuit size is not bounded by polylog(1/ϵ) – it can be much higher as has been empirically shown in the literature for various real implementations.
>
> **C3. The technique used seems to be to use quantum units to mimic subfunctions in a manner quite similar to classical units, and does not provide much insight into why we should expect a 'quantum improvement' here.**
>
> Thanks for the comment. In *Sec 2.2 of Revision-1*, we have clarified that there emerging works focusing on implementing classical neural networks to quantum computing to make full use of high-parallelism of quantum computing and regard quantum computing as an accelerator. They have demonstrated competitive accuracy against the classical neural networks with reduced time complexity (Francesco et al, 2019; Jiang et al, 2020).
> In Sec 5 of Revision-1, we have discussed the insights obtained from the derived results and future works. Specifically, results in this work provide theoretical and practical insights into the design of neural networks for quantum computing to fully harvest the quantum power in the hybrid quantum-classical computing scheme.
> * Neural networks designed for hybrid quantum-classical computing, including TensorFlow Quantum, have the ability to approximate a wide class of functions with arbitrarily small error.
> * For the near-term hybrid quantum-classical neural network designs, the proposed prologue-acceleration-epilogue architecture is a promising computing scheme to achieve high accuracy with only two interfaces between quantum and classical portions for data conversion.
> * Based on the depth complexity of $O(1)$ for a hybrid quantum-classical neural network, it inspires that the design of neural network for quantum computing may consider a “shallow” quantum circuit, instead of the “deep” version on classical computers. The power of shallow circuits has been demonstrated in recent works.
>
> Putting all together, this work demonstrated the combination of machine learning and quantum computing is a promising research direction, and the results can guide future research works in the design of neural networks for quantum computing.

---

### Official Review · AnonReviewer2 · 2020-11-01
**hard to understand context and significance**

**Rating:** 6
**Confidence:** 2

**Review:**

This paper provides new approximation theorems for a family of functions representable by hybrid quantum-classical circuits.


Specifically, the paper looks at neural nets that can be evaluated quickly by a three-stage circuit, where the second stage is quantum and the first and third phases are classical. The paper shows that neural networks in such a class allow for more efficient approximation of certain functions than (a) was previously known and (b) is possible using a related class of networks that can be implemented using an entirely classical circuit.



Overall, I found the paper’s significance hard to evaluate. Here are a few questions I had a hard Time finding answers to in the text of the paper:

- Why is this problem important to begin with? Is the bottleneck with neural nets their evaluation, or their training? Is the hope that by using quantum computers to evaluate complex NN’s we can allow for learning more complex functions? How would one train such networks?  I felt like the big picture was missing. I know my way around quantum computing and algorithms but still found the paper hard to read and understand. I think a typical ICLR audience-member would be entirely lost.

- The comparison with known bounds for classical circuits in Section 5 was quite hard to understand. Taking it at face value, the gains in complexity from moving to the quantum model seem limited: gains of $poly\log(1/\epsilon)$ in depth and circuit size? Is it clear that no better classical circuits are known for approximating the functions of interest?

Some specific comments and suggestions:

- Equation 1: What does it mean that y is bounded when sigma is a polynomial? Does sigma have to be such that it’s value is bounded on the range of possible inputs (something like $-(d+1)$ to $(d_1)$)?
- Expand the notation in the definition of equation (2). I interpreted it to mean that derivatives of up to order n are defined almost everywhere, and bounded. But the comment after that (when n=1 and f is not differentiable) confused me.
- Lemma 3.3: how complex is the network as a function of delta? I had trouble mapping the form of $f_2$ into the form of the network. I guess it has something like depth 3 and a number of neurons equal to the number of possible $\mathbf{ k, v}$ pairs?
- I felt like what was missing at the end of Section 3.2 was a self-contained recap statement.
- Moving the main Theorem statements (4.5 and 4.6) earlier to the introduction (along with the comparison to bounds for classical circuits) would help. (That is, explaining the result top-down, rather than bottom-up.) The effort to keep everything in mind until I got to final statements exceeded my stack depth. I like to think that I would have had no trouble understanding a more self-contained exposition of the main results.

---

> ### Author Response · Authors · 2020-11-14
> **Response to AnonReviewer2 (Part 1)**
>
> We appreciate your valuable time to give us helpful comments. In Revision-1, we have carefully considered your comments and taken actions to clarify or address them. The detailed actions are reported as follows.
>
> **C1: Why the problem is important?**
>
> The importance of the problem was initially discussed in Section 2 of the paper. In *Sec 2.2 of Revision-1*, we have added additional discussion on the motivation of this work. Specifically, although there are increasing interests in quantum machine learning (like google Tensorflow Quantum and many academic research efforts) and the recent quantum neural networks have demonstrated competitive accuracy against the classical neural networks with reduced time complexity, a fundamental question remains unknown: there are existing theories that show neural networks on classical computers have universal approximability, i.e., they can approximate a wide class of functions with arbitrarily small error, but do neural networks on quantum computers have the same universal approximability? This question has risen from the fact that the quantum circuit has limitations in computation operations to achieve high parallelism, such that classical neural networks cannot be directly mapped to quantum circuits. For example, quantum circuits do not allow common activation functions, like ReLU and Sigmod. This question is important since the failure to attain universal approximability will fundamentally hinder quantum neural networks from achieving state-of-the-art performance on various machine learning tasks, say image classification.
>
> **C2: Readability problem caused by the organization.**
>
> In Revision-1, we have followed your suggestion to move the main results to *Section 3* to organize the whole paper in a top-down structure. Thanks for the suggestion. In addition, we have added examples and discussions to make the paper easy to follow.
>
> **C3: What is the hope of using quantum computers for NN? What is the efficiency of quantum neural networks for evaluation and training?**
>
> In general, quantum computing has the potential to accelerate the neural networks for both evaluation and training with proper design, as has been shown in various literature already. But it is not the goal of this work. We have clarified in *Sec 2.2 and Sec 3 of Revision-1*. In this paper, we are trying to explore a more fundamental theoretical problem: due to the limitation in quantum computing scheme, can neural networks still have the universal approximability when they are running on quantum computers?  Following similar paths how the literature demonstrates the universal approximability of neural networks on classical computers, in this paper, we show that neural networks as function approximators can also achieve universal approximability on quantum computers. This provides theoretical support to continue the exploration of the implementation of neural networks on quantum computers.
>
> **C4: Comparison of known bounds.**
>
> First, the primary goal of this paper is to prove the universal approximability of neural networks on quantum computers, a missing piece in the literature. Based on the constructed network, we further give the complexity bound analysis. Our conclusion is that neural networks on quantum computers can achieve a lower depth upper bound and smaller circuit size compared with those on classical computers. But it is in big O notation, so the actual reduction in depth and circuit size is not bounded by polylog(1/ϵ) – it can be much higher as has been empirically shown in the literature for various real implementations.  Second, these bounds used in the comparison are recently developed for neural networks on classical computers. We are not aware of better bounds in the literature.
>
> **C5: Question on Equation 1.**
>
> We have clarified in *Sec 4.1 of Revision-1* that the network requires that the square of the sum of all inputs to be normalized to 1, which is the requirement by using quantum states for computation. As a result, y will be in [-1, 1] after an nth-order polynomial function of sigma, where $n\ge 1$.
>
> **C6: Question on Equation 2.**
>
> In *Sec 4.2 of Revision-1*, we have clarified that the notation $D^{n}f(x)$ means the weak derivative (not strong derivative) of up to order n. Therefore, the function does not need to be differentiable. We have removed the confusing statement of $n=1$.

---

> > ### Author Response · Authors · 2020-11-14
> > **Response to AnonReviewer2 (Part 2)**
> >
> > **C7. Question on Lemma 3.3.**
> >
> > First, in *Sec A2 of Revision-1*, we have clarified that the complexity of the neural network build to obtain f2 is related to the possible k,v pairs. More specifically, the number of terms is $(S+1)^d(\frac{d^n-1}{d-1})$. For the implementation of each term in BPNN, it involves the multiplication of a constant and an input variable with the $n^{th}$ order non-linear function, which can be implemented using the basic neural operations in Figure 2 with a constant number of weights (i.e., 2 for constant generation and 6 for multiplication). Then, all the terms will be sum up using a linear activation function, and the number of weights is the same as the number of terms. As a result, the complexity of the corresponding neural network is $O(S^dd^{n-1})$.
> >
> > In *Sec 4.2 of Revision-1*, we have added an example using Figure 2 to demonstrate how to map the form of f2 in Lemma 4.3 into the form of a network.
> >
> > **C8. Self-contained recap statement of Sec 3.2.**
> >
> > Thanks for the suggestion. In *Sec 4.2 of Revision-1*, we have added the recap statement at the end of this section.

---

### Author Response · Authors · 2020-11-14
**Summary of Changes of Revision-1**

We would like to express our sincere thanks to the reviewers for their valuable time and constructive comments to help us to improve the quality of this paper. We are glad to report that this revision has complied with all the review comments. We denote this version of revision as "Revision-1".

In the responses, we have listed the detailed actions for each comment. The review comments are shown in **Bold font**, our detailed responses are shown in "Normal font", and the position of modifications in Revision-1 using *Italic font*. In Revision-1, we have highlighted the modified contents using "red color".

#### Summary of Changes:
* (Section 2) We have added one set of experimental results to demonstrate that the current state of implementing neural networks on quantum accelerators. In addition, we have added more discussions on the motivation of this work and emphasized the importance of the study on the proof of the universal approximability of neural networks design for hybrid quantum-classic computing; in particular the TensorFlow Quantum.
* (Section 3) We have added a new section by moving the main results to this section. We have also clarified that the proposed prologue-acceleration-epilogue computing scheme is a special case of hybrid quantum-classical computing used in TensorFlow Quantum, which implies that if we prove the neural networks designed for the target computing scheme having the universal approximability, the conclusion can be held for the wide hybrid quantum-classical computing platform, like TensorFlow Quantum.
* (Section 4) We have rewritten the function space to make it clearer and added a recap statement after Sec 4.2. In addition, we have provided examples to explain the operations in the designed quantum circuits.
* (Section 5) We have added the practical insights derived from the main results obtained in this work and discussed how the findings in this work can be used for future research in the quantum machine learning field.
* (Reference) We have added 8 new related references.
  * [1] Marcello Benedetti, Delfina Garcia-Pintos, Oscar Perdomo, Vicente Leyton-Ortega, Yunseong Nam, and Alejandro Perdomo-Ortiz. A generative modeling approach for benchmarking and training shallow quantum circuits. npj Quantum Information, 5(1):1–9, 2019.
  *[2] Marco Cerezo, Akira Sone, Tyler Volkoff, Lukasz Cincio, and Patrick J Coles. Cost-function-dependent barren plateaus in shallow quantum neural networks. arXiv preprint arXiv:2001.00550,2020.
  *[3] Iris Cong, Soonwon Choi, and Mikhail D Lukin.  Quantum convolutional neural networks. Nature Physics, 15(12):1273–1278, 2019.
  *[4] Alberto Delgado. Function Approximation with Quantum Circuit. 2018.
  *[5] William Huggins, Piyush Patil, Bradley Mitchell, K Birgitta Whaley, and E Miles Stoudenmire. Towards quantum machine learning with tensor networks. Quantum Science and technology, 4(2):024001, 2019.
  *[6] Yann LeCun, L ́eon Bottou, Yoshua Bengio, and Patrick Haffner. Gradient-based learning applied to document recognition. Proceedings of the IEEE, 86(11):2278–2324, 1998.
  *[7] Alejandro Perdomo-Ortiz, Marcello Benedetti, John Realpe-G ́omez, and Rupak Biswas. Opportunities and challenges for quantum-assisted machine learning in near-term quantum computers. Quantum Science and Technology, 3(3):030502, 2018.
  *[8] Rongxin Xia and Sabre Kais.  Hybrid Quantum-Classical Neural Network for Calculating Ground State Energies of Molecules. Entropy, 22(8):828, 2020.

---

### Decision · Program_Chairs · 2021-01-07
**Final Decision**

**Decision:**

Reject

**Comment:**

This paper provides approximation results for functions that can be represented by hybrid quantum-classical circuits. It is felt that venues such as QIP would be a more suitable venue, and perhaps some experiments/simulations could be added.